# Towards Personalized Privacy: User-Governed Data Contribution for Federated Recommendation

## ABSTRACT

Federated recommender systems (FedRecs) have gained significant attention for their potential to protect user privacy by keeping user privacy data locally and only communicating model parameters/gradients to the server. Nevertheless, the currently existing architecture of FedRecs assumes that all users have the same 0-privacy budget, i.e., they do not upload any data to the server, thus overlooking those users who are less concerned about privacy and are willing to upload data to get a better recommendation service. To bridge this gap, this paper explores a user-governed data contribution federated recommendation architecture where users are free to take control of whether they share data and the proportion of data they share to the server. To this end, this paper presents a cloud-device collaborative graph neural network federated recommendation model, named CDCGNNFed. It trains user-centric ego graphs locally, and high-order graphs based on user-shared data in the server in a collaborative manner via contrastive learning. Furthermore, a graph mending strategy is utilized to predict missing links in the graph on the server, thus leveraging the capabilities of graph neural networks over high-order graphs. Extensive experiments were conducted on two public datasets, and the results demonstrate the effectiveness of the proposed method.

## CCS CONCEPTS

• **Information systems** → **Recommender systems**.

## KEYWORDS

Federated learning, recommender systems, user-controlled learning

**ACM Reference Format:**
Anonymous Author(s) . 2018. Towards Personalized Privacy: User-Governed Data Contribution for Federated Recommendation. In *Proceedings of Make*

*sure to enter the correct conference title from your rights confirmation emai (Conference acronym 'XX)*. ACM, New York, NY, USA, 9 pages. https://doi.org/XXXXXXX.XXXXXXX

## 1 INTRODUCTION

Recommender systems [3, 35] have been shown to be an effective technique for providing personalised content recommendation services (e.g., videos and goods) to users based on their preferences. Typically, the recommender system is deployed on a central server that collects all users' historical behavior data (e.g., clicks and purchases) to train a global recommendation model, and the more data that is collected, the more accurate the model is. However, such recommender systems inevitably raise privacy concerns due to their centralized data collection mechanism. Moreover, many regulations, such as the General Data Protection Regulation (GDPR[1]), have recently been issued to better protect users' data privacy, so it is desirable to investigate how to balance privacy risks against recommendation utilities.

Recently, federated learning [22, 31], as a promising solution to privacy-preserving machine learning, has been widely adopted in recommender systems to mitigate privacy concerns, termed federated recommender systems *(FedRec)* [25, 30]. Specifically, as shown in Figure 1 (a), the key idea of FedRec is that all of the user's data is retained on their own device in a decentralized fashion. In each training round, the central server randomly selects a group of devices to train locally, and then only parameters/gradients (without actual data sharing) are aggregated to the central server to learn a global model that will be redistributed to each device. Research on FedRecs could be roughly classified into two categories: matrix factorization based FedRecs (MF-FedRecs) [1, 4, 19] and graph neural networks based FedRecs (GNN-FedRecs) [20, 21, 27]. MF-FedRecs mainly learns the global item embedding table by collaboratively training the local first-order user-item interaction matrix distributed across different devices. On the other hand, GNN-based recommender systems [29] have recently achieved state-of-the-art results due to their superior ability to effectively capture higher-order graph structural information compared to matrix factorization methods. However, in federated scenarios, each device has only a first-order user ego graph that includes the items that the user interacts with directly. Hence, the core challenge behind GNN-FedRecs is to learn higher-order graph structural information in a privacy-preserving manner. For example, FedGNN [27] presents to use a trusted third-party server to construct a high-order graph.

---

[1]https://gdpr-info.eu/

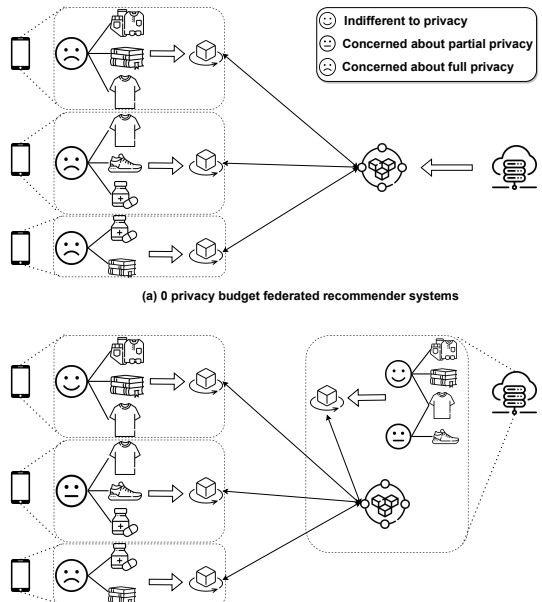


Indifferent to privacy
Concerned about partial privacy
Concerned about full privacy

(a) 0 privacy budget federated recommender systems

(b) Personalized privacy budget federated recommender systems


**Figure 1: (a) 0 privacy budget federated recommender systems that users do not upload any data to the server. (b) Personalized privacy budget federated recommender systems that users are free to take control of whether they share data and the proportion of data they share to the server.**

Nevertheless, the currently existing architecture of FedRecs assumes that all users have the same 0-privacy budget, meaning they do not upload any data to the server, which is inflexible and unappealing due to the following reasons: (1) it overlooks those users who are less concerned about privacy and willing to share either all or portions of their data to receive a better recommendation service. (2) The model performance of FedRec is generally degraded due to the non-identically distributed data among the users' devices [13, 17]. Thus, adopting a uniform policy where all users are prohibited from uploading data could also hurt the revenue of platforms due to the degraded model performance. (3) It requires that users who need the recommendation service have to be involved in model training, which brings a huge burden to the user's device as it requires substantial computational and storage resources as well as communication costs.

To mitigate above issues, this paper explores a user-governed data contribution federated recommendation architecture where users are free to take control of whether they share data and the proportion of data they share to the server. In such a setting, this paper presents a cloud-device collaborative graph neural network federated recommendation model, named CDCGNNFed. It trains user-centric ego graphs locally, and high-order graphs based on user-shared data in the server in a collaborative manner via contrastive learning. Specifically, a graph mending strategy is first employed to predict missing links in the graph on the server, thus leveraging the capabilities of graph neural networks over high-order graphs. After that, for each training round, devices and the

server independently infer and exchange embeddings, so that local and global views of the same node can be constructed as positive pairs for contrastive learning.

Overall, our main contributions are summarized as follows:

- To our best knowledge, this is the first work to investigate a more flixible and personalized privacy framework called user-governed data contribution federated recommendation (UGFedRec), where users have granular control over the extent to which they are willing to share data with the platform to balance privacy risks and recommendation utilities.
- In the UGFedRec setting, we propose a cloud-device collaborative graph neural network federated recommendation model, named CDCGNNFed, which trains user-centric ego graphs locally, and high-order graphs based on user-shared data in the server in a collaborative manner via contrastive learning.
- We conduct extensive experiments on public real-world datasets to validate the effectiveness of the proposed methods, and experimental results demonstrate that the proposed method can achieve a promising performance for the Top-K recommendation.

The remainder of this paper is organized as follows. Section 2 will review related work, and Section 3 will formulate the research problem and elaborate on the proposed method. The experiments are discussed in Section 4, followed by a conclusion in Section 5.

## 2 RELATED WORK

### 2.1 Centralized Recommendation

Recommender systems have been shown to be an effective technique for providing personalised content recommendation services (e.g., videos and goods) to users by collecting all users' historical behavior data (e.g., clicks and purchases) to train a global recommendation model on the server. Methods in this field can be broadly categorised as MF-based methods, deep learning based methods and GNN-based methods. The main idea of MF-based methods [15, 23] is to decompose the user-item interaction matrix into two lower-dimensional matrices representing latent features of users and items. Deep learning based methods [6, 8, 32] focus on leveraging deep neural networks to learn intricate patterns from user-item interaction data, often capturing non-linear relationships. In recent years, GNN-based recommender systems [10, 29, 33] have achieved state-of-the-art results due to their superior ability to effectively capture higher-order graph structural information. As previously highlighted, these methods are largely centralized, collecting user data for model training, which raises potential data privacy concerns.

### 2.2 Federated Recommendation

Drawing inspiration from the efficacy of federated learning in ensuring privacy in machine learning, FedRecs have been introduced, allowing for cloud-device model collaborative training without actual data sharing. Research on FedRecs could be roughly classified into two categories: matrix factorization based FedRecs (MF-FedRecs) [1, 4, 19] and graph neural networks based FedRecs (GNN-FedRecs) [20, 21, 27]. MF-FedRecs mainly learns the global item

embedding table by collaboratively training the local first-order user-item interaction matrix distributed across different devices. For example, FCF [1] extends collaborative centralized filtering to the federated model. In particular, it utilizes alternating least squares and stochastic gradient descent to optimize user and item embeddings on the device and server sides, respectively. On the other hand, GNN-based recommender systems [29] have recently achieved state-of-the-art results due to their superior ability to effectively capture higher-order graph structural information compared to matrix factorization methods. For instance, FedGNN [27] presents to use a trusted third-party server to construct high-order graph such that GNNs could be employed to learn user/item embeddings in a privacy-preserving manner. Although currently FedRecs have attracted considerable interest in the privacy-preserving recommendation field, methods in the context of user-governed data contribution federated recommendation remain highly unexplored. A work similar to us is FedeRank [2], where users also have the ability to govern the proportion of data they upload. However, a distinguishing factor from our method is that in Federank, all user data remains local, while users can dictate the percentage of gradients corresponding to training samples that they transmit.

## 3 PROPOSED METHOD

In this section, we first formulate the research problem and then elaborate the proposed method.

### 3.1 Problem formulation

Let $\mathcal{U}$ and $\mathcal{I}$ represent a set of users/devices[2] and items, respectively. $X \in \mathbb{R}^{|\mathcal{U}| \times |\mathcal{I}|}$ denotes the binary user-item interaction matrix where the element $x_{ui}$ represents the implicit feedback between user $u \in \mathcal{U}$ and item $i \in \mathcal{I}$. Specifically, $x_{ui} = 1$ and $x_{ui} = 0$ indicate whether there is an interaction or not, respectively. In addition, the embedding-based recommendation model is denoted as $f(\Theta)$ parameterized by $\Theta$, such as matrix factorization based methods and graph neural network based methods. Specifically, it maps users and items into a shared embedding space via the model $f(\Theta) : \mathcal{U}, \mathcal{I} \rightarrow P \in \mathbb{R}^{|\mathcal{U}| \times d}, Q \in \mathbb{R}^{|\mathcal{I}| \times d}$, where the user embedding $\mathbf{p}_u \in P$ and the item embedding $\mathbf{q}_i \in Q$ represent the $d$-dimensional vector representations of the user $u$ and the item $i$, respectively.

In the traditional federated recommendation setting, each user $u$ keeps all of their own interaction data $X_u \in \mathbb{R}^{|\mathcal{I}|}$ that correspond to the $u$-th row of $X$ on their local device for the purpose of privacy protection. In addition, each device $u$ maintain its local model consisting of a set of local parameters that have model parameters $\Theta_u$, the user embedding $\mathbf{p}_u \in \mathbb{R}^d$, and the item embedding table $Q_u \in \mathbb{R}^{|\mathcal{I}| \times d}$. For each training round, the server selects a set of devices, denoted $\mathcal{U}_s \subseteq \mathcal{U}$, to train their models locally. Each local device typically uploads the locally trained parameters $\Theta_u$ and the item embedding table $Q_u$ or their corresponding gradients $\nabla \Theta_u$ and $\nabla Q_u$ to the server. After that, the server will train a global model based on the collected parameters/gradients from $\mathcal{U}_s$ via the aggregation function, such as FedAvg [22], and then redistributes the global model to all devices.

---

[2]We assume that each device is only associated with a single user. Therefore, we interchangeably use the terms "device" and "user" throughout this paper.

Although the federated recommendation architecture above can protect users' privacy by keeping all users' data locally, it assumes that all users have the same 0-privacy budget, i.e., they do not upload any to the server, thus overlooking those users (denoted $\mathcal{U}^+$) who are less concerned about privacy and willing to share either all or portions of their data to receive a better recommendation service. To bridge this gap, this work aims to explore a more flexible federated recommendation framework, termed user-governed data contribution federated recommender system (UGFedRec), where users are free to take control of whether they share data and the proportion of data they share with the server. Specifically, the main difference between UGFedRec and traditional FedRec is that each user has the option of uploading a certain percentage of the interaction data $X_u^+ \in \mathbb{R}^{|\mathcal{I}|'}$ to the server, where $0 \leq |\mathcal{I}|' \leq |\mathcal{I}|$ is the number of data users are willing to upload to the service. In this way, the server can also train a cloud model consisting of model parameters $f_s(\Theta_s)$ and the item embedding table $Q_s \in \mathbb{R}^{|\mathcal{I}| \times d}$ based on the data $X^+ \in \mathbb{R}^{|\mathcal{U}^+| \times |\mathcal{I}|}$, where $|\mathcal{U}^+| \leq |\mathcal{U}|$ is the number of users willing to upload data. Finally, the goal of UGFedRec is to minimize the following loss function $\mathcal{L}$:

$$\mathcal{L} = \sum_{u \in \mathcal{U}} \mathcal{L}_u + \mathcal{L}_s \tag{1}$$

where $\mathcal{L}_u$ and $\mathcal{L}_s$ are loss functions for locel devices and the server, respectively.

### 3.2 CDCGNNFed

This work aims to explore a more flexible and personalized privacy framework for user-governed data contribution federated recommendation (UGFedRec). To this end, we introduce a cloud-device collaborative graph neural network federated recommendation model, named CDCGNNFed. The architecture of the proposed method is shown in Figure 2, we first simulate that users can freely choose what percentage of historical interaction data to share with the cloud server, so that the device-side model and the cloud-side model can train models based on local data and data that users are willing to share, respectively. To leverage the capabilities of graph neural networks over high-order graphs, we introduce a graph mending strategy to predict missing links in the graph on the server. After that, the server randomly select a group of devices for each training round, and they independently infer and exchange embeddings, so that local and global views of the same node can be constructed as positive pairs for contrastive learning. The detailed algorithm is described as Algorithm 1.

### 3.3 Graph mending

Within the context of UGFedRec, users voluntarily contribute either all or a portion of their interaction data to the server. Consequently, the server constructs a user-item bipartite graph $\mathcal{G} = \{\mathcal{U}^+, \mathcal{I}, \mathcal{E}^+\}$, where $\mathcal{U}^+$ is the set of users willing to share their data, $\mathcal{I}$ denotes the set of all items, and $\mathcal{E}^+ \subset \mathcal{U}^+ \times \mathcal{I}$ denotes the user-item interaction data provided by users. Nevertheless, when the volume of user-contributed data is limited, the graph $\mathcal{G}$ on the server may be disjointed, consisting of several subgraphs representing isolated user-item interactions. As a result, simply applying a GNN directly to these subgraphs might not effectively capture higher-order graph

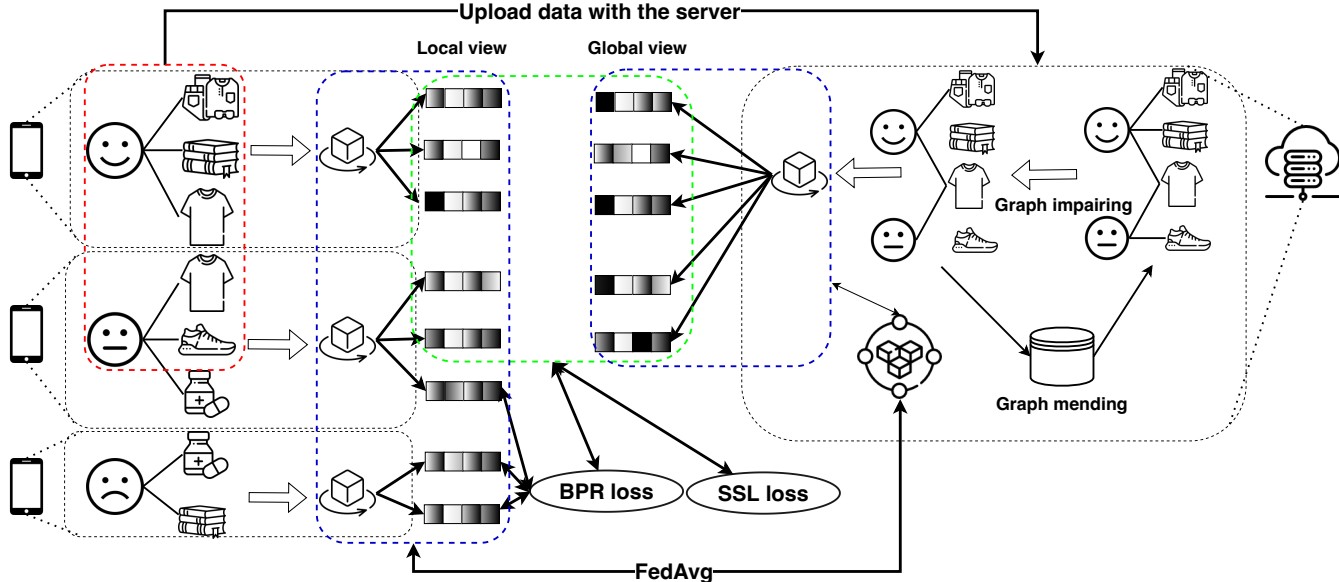

**Figure 2: The architecture of the proposed method. Users can choose to upload all, some, or no data to the server. The server employs a graph mending strategy to predict missing links, and subsequently, both the server and local devices collaboratively learn user/item embedding in a contrastive learning fashion.**

structural information, potentially leading to suboptimal outcomes. To address this challenge, inspired by the approach in FedSage+ [34], we introduce a graph mending strategy that predicts the missing links in graph $\mathcal{G}$. This enables the GNN to fully exploit its node representation capability, ensuring more comprehensive information capture from the graph structure. Specifically, we first employ a graph impairing strategy to extract or "impart" a subset of links $\hat{\mathcal{E}^+} \subset \mathcal{E}^+$ serving as the ground truth for training graph mending by simulating the scenario wherein links may be missing. With the impaired links in place, we can leverage standard GNNs for learning the node embedding over graph $\mathcal{G}$ as follows:

$$Z_u, Z_i = GNN(\mathcal{G}) \tag{2}$$

where $\mathbf{z}_u \in Z_u, \mathbf{z}_i \in Z_i$ are the user embedding and the item embedding, respectively. $GNN(\cdot)$ denotes GNN-based node encoder models. Since $\mathcal{G}$ is a bipartite graph in this context, we adopt Light Graph Convolution (LGC [11]) as the encoder, which will be provided more detailed introduction in the subsequent sections. In this way, for the pair of user-item nodes involved in the impaired link $e_{ui} \in \hat{\mathcal{E}^+}$, we can use cosine similarity (denoted $cos(\cdot)$) to calculate the distance between the pair of nodes, and update the user/item embeddings using the following loss function:

$$\mathcal{L}_{gm} = \sum_{e_{ui} \in \hat{\mathcal{E}^+}} ||cos(\mathbf{z}_u, \mathbf{z}_i) - e_{ui}||_2, e_{ui} = 1, e_{ui} \in \hat{\mathcal{E}^+}; e_{ui} = 0, e_{ui} \notin \hat{\mathcal{E}^+}$$

$$\tag{3}$$

Finally, we can predict the potential missing links by calculating the cosine similarity of user-item node pairs of $\mathcal{G}$, and determine whether to establish a link by comparing it to a predefined threshold $t$.

### 3.4 Embedding inference

After predicting the missing links on the server side, we perform embedding inference separately on the device side and server side, aiming to obtain the local view of the same node on the device side and the global view on the server side.

*3.4.1 Device side embedding inference.* Since each device has only a first-order user ego graph that includes the items that the user interacts with directly. Thus, the user/item embeddings could be learned by LGC [11] as follows:

$$\mathbf{e}_{u-d}^{(l+1)} = \sum_{i \in \mathcal{N}_{(u)}} \frac{1}{\sqrt{|\mathcal{N}_{(u)}|}\sqrt{|\mathcal{N}_{(i)}|}} \mathbf{e}_{i-d}^{(l)}$$

$$\mathbf{e}_{i-d}^{(l+1)} = \sum_{u \in \mathcal{N}_{(i)}} \frac{1}{\sqrt{|\mathcal{N}_{(i)}|}\sqrt{|\mathcal{N}_{(u)}|}} \mathbf{e}_{u-d}^{(l)}$$

$$\tag{4}$$

where $\mathbf{e}_{u-d}$ and $\mathbf{e}_{i-d}$ denote the user embedding and the item embedding derived from the device side, respectively.

*3.4.2 Server side embedding inference.* Since the graph on the server side possesses higher-order graph structural information after graph mending, we first use the same method as in Equation (4) to calculate the embeddings of the user and item nodes at each layer. We then use the following layer combination method [11] to obtain the final user/item embeddings on the server side, denoted $\mathbf{e}_{u-s}$ and $\mathbf{e}_{i-s}$, respectively.

$$\mathbf{e}_{u-s} = \sum_{l=0}^{L} \alpha_l \mathbf{e}_{u-s}^{(l)}; \quad \mathbf{e}_{i-s} = \sum_{l=0}^{L} \alpha_l \mathbf{e}_{i-s}^{(l)} \tag{5}$$

where $\alpha_l$ is the hyperparameter representing the importance of the $l$-th layer.

## 3.5 Device-cloud constrastive learning

After obtaining the user/item embeddings on both the local device and the server, we allow those devices willing to upload data to the server to exchange their corresponding user/item embeddings with the server. As a result, for the same user/item node, we can consider the embedding obtained from the local device as the node's local view, and the embedding from the server as its global view. In this way, we can construct a positive pair, that is, $\{(\mathbf{e}_{u-d}, \mathbf{e}_{u-s})|u \in \mathcal{U}^+\}$ for contrastive learning, and treat views from different nodes as negative pairs, that is, $\{(\mathbf{e}_{u-d}, \mathbf{e}_{v-s})|u,v \in \mathcal{U}^+, u \neq v\}$. Formally, we follow SimCLR [5] to adopt contrastive loss InfoNCE [9] as follows:

$$\mathcal{L}_{CL}^{user} = \sum_{u \in \mathcal{U}^+} -log\frac{exp(cos(\mathbf{e}_{u-d}, \mathbf{e}_{u-s})/\tau)}{\sum_{v \in \mathcal{U}^+} exp(cos(\mathbf{e}_{u-d}, \mathbf{e}_{v-s})/\tau)} \quad (6)$$

where $\tau$ is the hyperparameter known as temperature. Analogously, the contrastive loss of items is denoted as $\mathcal{L}_{CL}^{item}$. In this way, the final self-supervised loss is $\mathcal{L}_{SSL} = \mathcal{L}_{CL}^{user} + \mathcal{L}_{CL}^{item}$ [28].

## 3.6 Local and global model update

In the context of local model updates for devices, for users who are not willing to upload their data to the server, we leverage the data available on their local devices and update using the Bayesian Personalized Ranking (BPR) [24] loss function $\mathcal{L}_{BPR}$ as follows:.

$$\mathcal{L}_{BPR} = -\sum_{i \in \mathcal{N}_u} \sum_{j \notin \mathcal{N}_u} \ln \sigma \left(cos(\mathbf{e}_{u-d}, \mathbf{e}_{i-d}) - cos(\mathbf{e}_{u-d}, \mathbf{e}_{j-d})\right) + \lambda \|\mathbf{e}\|^2 \quad (7)$$

where $\lambda$ is a hyperparameter for controlling the strength of the $L_2$ regularization. Analogously, for those users who willing to share their data with the server and for the server-side model, we train models using a combination of the BPR loss and self-supervised loss, as illustrated below.

$$\mathcal{L} = \mathcal{L}_{BPR} + \lambda_1 \mathcal{L}_{SSL} + \lambda_2 \|\mathbf{e}\|^2 \quad (8)$$

where $\lambda_1$ and $\lambda_2$ are hyperparameters for controlling the strength of self-supervised loss and regularization. Finally, we can employ various parameter aggregation methods, such as FedAvg [22], to update the global parameters and then redistribute to all devices.

## 4 EXPERIMENTS

In this section, we will first introduce the experimental settings, and then report and discuss the experimental results for answering the following research questions:

- How does the proposed method compare with other federated recommendation methods in the UGFedRec setting?
- How do various components, such as contrastive learning, influence the performance of the proposed method?
- How do various hyperparameters influence the performance of the proposed method?

## 4.1 Settings

*4.1.1 Datasets.* To validate the effectiveness of the proposed method, the experiment is carried out on two public datasets, including Gowalla [18], and Yelp2018[3]. The statistics of datasets are described

[3]https://www.yelp.com/dataset/challenge

---

**Algorithm 1** CDCGNNFed

---

1: Users voluntarily contributed data to the server to construct a graph $\mathcal{G} = \{\mathcal{U}^+, \mathcal{I}, \mathcal{E}^+\}$
2: Initialize model parameters $\Theta_u, \mathbf{p}_u, Q_u$ for each client and the server.
3: **for** t=0,... **do**                  ▷ For each training round
        // In the device side
4:     Select a set of devices $\mathcal{U}_s$ randomly
5:     **for** each device $u \in \mathcal{U}_s$ in parallel **do**
6:         Infer local user/item embedding via (4)
7:         Upload user/item embedding to the server
8:     **end for**
        // In the server side
9:     Predict missing links via Equation (3)
10:    Infer user/item embeddings via Equation (4) and (5)
11:    Distribute user/item embeddings to the device $u \in \mathcal{U}_s$
        // Device-cloud contrastive learning
12:    **for** each device $u \in \mathcal{U}_s$ in parallel **do**
13:        local model training via Equation (7) and/or (8)
14:        Upload local gradients to the server
15:    **end for**
16:    server model training via Equation (8)
        // Global model update
17:    Update global model via FedAvg
18:    Distribute global model to all devices
19: **end for**                           ▷ Until model convergence

---

in Table 1. The Gowalla is a location-based social network dataset consisting of users and their locations by checking-in. On the other hand, the Yelp2018 dataset is a business review dataset that includes customers, restaurants, and the associated reviews given by customers to these restaurants. Following [12, 26], we exclude users and items from Gowalla and Yelp2018 that have fewer than 20 and 10 interactions, respectively. Each of the three datasets is then partitioned into training, validation, and test sets in an 8:1:1 ratio, respectively.

**Table 1: The statistics of datasets.**

| Datasets | #Users | #Items | #Interactions |
|----------|--------|--------|---------------|
| Gowalla  | 29858  | 40981  | 1027370       |
| Yelp2018 | 31668  | 38048  | 1561406       |

*4.1.2 Baselines:*

- **Cloud-based recommendation methods:**
  - **NeuMF** [12]: It is the state-of-the-art MF-based deep recommendation method, utilizing DNN to supplant the dot product function, thereby capturing the non-linearity present in implicit feedbacks.
  - **LightGCN** [11]: It is the state-of-the-art GNN-based recommendation method, utilizing GNN to capture high-order graph structure information via the linear neighborhood aggregation mechanism.
- **FedRecs:**

**Table 2: The model performance with respect to Recall@20 and NDCG@20 on Gowalla for the partial uploading case.**

| Share ratio | | (0,0.1) | [0.1,0.2) | [0.2,0.3) | [0.3,0.4) | [0.4,0.5) | [0.5,0.6) | [0.6,0.7) | [0.7,0.8) | [0.8,0.9) | [0.9,1) |
|---|---|---|---|---|---|---|---|---|---|---|---|
| Recall@20 | FedeRank | 0.1438 | 0.1443 | 0.145 | 0.1453 | 0.1461 | 0.1468 | 0.1474 | 0.1479 | 0.1484 | 0.1491 |
| | UGFed-MF | 0.1442 | 0.145 | 0.145 | 0.1463 | 0.1471 | 0.1478 | 0.1486 | 0.1493 | 0.15 | 0.1502 |
| | UGFed-GNN | 0.1453 | 0.146 | 0.1468 | 0.1498 | 0.1553 | 0.1573 | 0.1618 | 0.1704 | 0.1705 | 0.1752 |
| | **CDCGNNFed** | **0.1463** | **0.1478** | **0.1502** | **0.1531** | **0.1556** | **0.1583** | **0.1704** | **0.1727** | **0.1778** | **0.1809** |
| NDCG@20 | FedeRank | 0.1206 | 0.1215 | 0.1228 | 0.1236 | 0.1247 | 0.1253 | 0.1268 | 0.1274 | 0.1287 | 0.1294 |
| | UGFed-MF | 0.1227 | 0.1235 | 0.1241 | 0.125 | 0.1258 | 0.1263 | 0.1272 | 0.128 | 0.1295 | 0.1302 |
| | UGFed-GNN | 0.1163 | 0.119 | 0.1216 | 0.1245 | 0.1272 | 0.1301 | 0.1328 | 0.1356 | 0.1383 | 0.141 |
| | **CDCGNNFed** | **0.1228** | **0.1236** | **0.1244** | **0.1278** | **0.131** | **0.1345** | **0.1379** | **0.1452** | **0.1481** | **0.154** |

**Table 3: The model performance with respect to Recall@20 and NDCG@20 on Yelp 2018 for the partial uploading case.**

| Share ratio | | (0,0.1) | [0.1,0.2) | [0.2,0.3) | [0.3,0.4) | [0.4,0.5) | [0.5,0.6) | [0.6,0.7) | [0.7,0.8) | [0.8,0.9) | [0.9,1) |
|---|---|---|---|---|---|---|---|---|---|---|---|
| Recall@20 | FedeRank | 0.0482 | 0.0491 | 0.0498 | 0.0506 | 0.0511 | 0.0521 | 0.0527 | 0.0533 | 0.0541 | 0.0562 |
| | UGFed-MF | 0.0486 | 0.0493 | 0.0498 | 0.0504 | 0.0512 | 0.0521 | 0.0529 | 0.0542 | 0.0556 | 0.0571 |
| | UGFed-GNN | 0.0461 | 0.0475 | 0.0489 | 0.051 | 0.0523 | 0.0541 | 0.0558 | 0.0574 | 0.0597 | 0.0615 |
| | **CDCGNNFed** | **0.0493** | **0.0505** | **0.0518** | **0.0531** | **0.0544** | **0.0557** | **0.0568** | **0.0582** | **0.0605** | **0.0623** |
| NDCG@20 | FedeRank | 0.041 | 0.0414 | 0.0419 | 0.0423 | 0.0428 | 0.0433 | 0.0438 | 0.0444 | 0.0451 | 0.046 |
| | UGFed-MF | 0.0415 | 0.042 | 0.0427 | 0.0432 | 0.0429 | 0.0435 | 0.0442 | 0.0448 | 0.0458 | 0.0467 |
| | UGFed-GNN | 0.0423 | 0.0432 | 0.0442 | 0.0454 | 0.0462 | 0.0475 | 0.0484 | 0.0495 | 0.0497 | 0.045 |
| | **CDCGNNFed** | **0.0432** | **0.0444** | **0.045** | **0.0461** | **0.0468** | **0.0479** | **0.0485** | **0.0498** | **0.0507** | **0.0515** |

**Table 4: The model performance with respect to Recall@20 and NDCG@20 on Gowalla and Yelp 2018 for the no uploading and full uploading cases.**

| Datasets | | Gowalla | | | | Yelp2018 | | | |
|---|---|---|---|---|---|---|---|---|---|
| | | Recall@20 | | NDCG@20 | | Recall@20 | | NDCG@20 | |
| Share ratio | | 0 | 1 | 0 | 1 | 0 | 1 | 0 | 1 |
| Cloud | NeuMF | - | 0.1509 | - | 0.1309 | - | 0.0586 | - | 0.0476 |
| | LightGCN | - | 0.1811 | - | 0.1534 | - | 0.0627 | - | 0.0509 |
| FedRec | FedMF | 0.1435 | - | 0.122 | - | **0.0482** | - | **0.0413** | - |
| | FedPerGNN | **0.1442** | - | **0.1233** | - | 0.0453 | - | 0.0409 | - |
| UGFedRec | FedeRank | 0.1432 | 0.1494 | 0.1197 | 0.1308 | 0.0476 | 0.0574 | 0.0408 | 0.047 |
| | UGFed-MF | 0.1435 | 0.1504 | 0.122 | 0.1312 | **0.0482** | 0.0583 | **0.0413** | 0.0475 |
| | UGFed-GNN | 0.1428 | 0.1776 | 0.1117 | 0.1538 | 0.0457 | 0.0626 | 0.0412 | 0.0503 |
| | **CDCGNNFed** | 0.1428 | **0.1823** | 0.1117 | **0.1553** | 0.0457 | **0.0639** | 0.0412 | **0.0522** |

– **FedMF** [4]: It is a MF-FedRec method which introduces a user-centric distributed matrix factorization framework, leveraging the homomorphic encryption technique to ensure users' privacy.

– **FedPerGNN** [27]: It is GNN-FedRec method which employs a trusted third-party server to allocate neighbors, who share co-interacted items, to individual users, thereby leveraing the capabilities of GNN on capturing the high-order graph information.

• **UGFedRecs**

– **FedeRank** [2]: It is a MF-based UGFedRec method. In contrast to our approach, this method still retains all user data locally. However, it allows users to control the proportion of gradients corresponding to the training samples that are uploaded to the server.

– **UGFed-MF, and UGFed-GNN:** In the UGFedRec setting, the most naive approach would be to treat the server, collecting data voluntarily uploaded by users, as another device equivalent to other user devices, and then proceed with standard federated learning. Thus, we adopt MF and GNN as base models respectively, denoted as UGFed-MF

and UGFed-GNN, to serve as baselines under this setting. We consider items that a user hasn't interacted with as potential candidates and report the results averaged across all users.

*4.1.3  Evaluation Metrics.* To evaluate the model performance, we employ two commonly used metrics, i.e., Recall@20 and NDCG@20 (Normalized Discounted Cumulative Gain) throughout experiments [11, 12]. The former measures the proportion of relevant items found within the top-20 recommendations, and the latter evaluates not only the presence of relevant items in the top-20 but also their ranking quality, with higher positions being more valuable. Following [11], we consider items that a user hasn't interacted with as potential candidates and report the results averaged across all users.

*4.1.4  Hyper-parameter Settings.* We employ Xavier method [7] to initialize user and item embeddings with the embedding dimension 64 for all methods. We use Adam [14] as the optimizer, and the learning rate and weight decay are search from $\{0.001, 0.0005, 0.0001\}$ and $\{0.005, 0.0001, 0.0005, 0.00001\}$ via grid search, respectively. In addition, the number of devices sampled for each training round is 256 and 528 for MovieLens-1M and Yelp, respectively. The number of GNN layers for devices and the server models is 1 and 3, respectively. We will discuss settings of other hyperparameters in section 4.4. The baselines are implemented by the codes provided by the authors.

## 4.2  Top-K Recommendation (RQ1)

We first validate the efficacy of our method on the prevalent top-k recommendation task commonly seen in recommendation systems. Our evaluation initially simulates scenarios where users have autonomy over data uploading, encompassing three distinct cases: (1) No uploading (i.e., sharing ratio of 0), which renders the model equivalent to a traditional federated recommendation system with a 0-privacy budget; (2) Partial uploading (with a sharing ratio between 0 and 1). Here, we employ uniform sampling to randomly designate an upload ratio for each user; (3) Full uploading (sharing ratio of 1), under which circumstance the model aligns with a centralized recommendation system. For each case, we independently execute the model five times using different random seeds and report the averaged outcomes, and all results are statistically significant with $p < 0.05$. The results for the partial data uploading scenario are delineated in Tables 2 and 3 for two distinct datasets, while results for no uploading and full uploading scenarios are reported in Table 4. From the results, we can observer that:

- Overall, our proposed method outperforms baselines in the majority of cases, attesting to the effectiveness of the approach we've introduced.
- In most cases, GNN-based methods outperform those built on MF. Notably, for partial uploaded scenarios with smaller sharing ratios, the advantage of GNN is less pronounced. This can possibly be attributed to the server collating predominantly independent lower-order graphs, thereby mitigating GNN's potential in capturing higher-order graph structures. However, as the sharing ratio increases, the GNN-based techniques significantly overshadow MF-based methods.

- For the partial uploading case, our method, in tandem with UGFed-MF and UGFed-GNN, frequently outperforms Federank. This observation is plausible since, unlike Federank, we directly upload data to the server, rather than transmitting select gradients.
- In the no uploading scenarios, our model's performance aligns closely with the standard federated model. The MF-based approaches yield relatively better results. This behavior is understandable as there is no supplementary data available for utilization, leading our system to revert to the conventional federated recommendation model. Conversely, the GNN-based method, which can only harness first-order graph information, results in a somewhat suboptimal outcome.
- For full uploading scenarios, our model exhibits superior performance compared to centralized approaches. We attribute this enhanced performance partly to the introduced contrastive learning component. Moreover, by predicting missing links, our method partly alleviates the data sparsity issue, which subsequently enhances model performance.

## 4.3  Ablation Study (RQ2)

In this section, we aim to demonstrate the effect of the graph mending strategy component for predicting missing links and the device-cloud constrastive learning component for learning local and global views for the same node. To this end, we implement CDCGNNFed without graph mending strategy and constrastive learning, denoted as *w/o GM* and *w/o CL*, respectively. The experiments are carried out on two datasets, and other settings are the same as the partial uploading scenario. Experimental results are reported in Table 5, from which we can observe that:

- Overall, the removal of any component results in a significant deterioration in model performance. This underscores the indispensability and efficacy of both components within the model.
- Notably, the removal of the GM component leads to a more pronounced degradation in performance for both models. A possible explanation for this is the sparsity of the two datasets. The GM component proves adept at predicting missing links, effectively mitigating the challenges posed by this sparsity.
- The performance also sees a marked decline upon the removal of the CL component. This reiterates the potency of leveraging the contrastive learning component in rendering the learned embeddings more expressive. Concurrently, it affirms that in the UGFedRec context, the naive approach of viewing the server as a specialized device is suboptimal.

## 4.4  Hyperparameter analysis (RQ3)

In this section, we investigate the impact of the four critical hyperparameters associated with our proposed method on the model's performance on Gowalla dataset. These hyperparameters include: (1) Threshold $t = \{0.2, 0.4, 0.6, 0.8\}$ for graph mending strategy. (2) Temperature $\tau = \{0.1, 0.2, 0.5, 1\}$ for constrastive learning; (3) The number of devices $|\mathcal{U}_s| = \{256, 528, 1024, 2048\}$ for each training

**Table 5: Ablation studies results with respect to GG and CL components.**

| Method | Gowalla | | Yelp2018 | |
| --- | --- | --- | --- | --- |
| | Recall@20 | NDCG@20 | Recall@20 | NDCG@20 |
| w/o GM | 0.1689 | 0.1438 | 0.052 | 0.0412 |
| w/o CL | 0.1702 | 0.1442 | 0.0524 | 0.0427 |
| CDCGNNFed | **0.1721** | **0.1446** | **0.0571** | **0.049** |

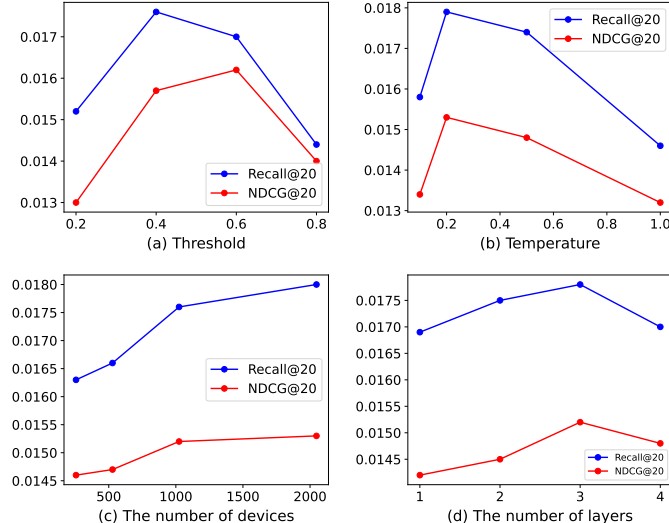

**Figure 3: The performance of model with different hyper-parameter settings on Gowalla dataset. (a) Threshold $t$; (b) Temperature $\tau$; (c) The number of devices $|\mathcal{U}_s|$; (d) The number of layers $l$.**

round; (4) The number of layers $l = \{1, 2, 3, 4\}$ for GNN. The experimental settings are the same as the partial uploading scenario, and results are shown in Figure 3. We can observe that:

- As the threshold increases, the model's performance initially rises but subsequently declines. This observation is rational, stemming from the fact that at lower thresholds, the graph mending strategy tends to produce a higher number of links, potentially introducing false negative links that degrade model performance. On the contrary, at higher thresholds, fewer links are generated, resulting in the persistence of numerous isolated subgraphs at the server end. This situation impedes the full exploitation of the Graph Neural Network's (GNN) inherent capabilities, leading to a drop in model performance.
- With regard to the temperature parameter in contrastive learning, the model's performance significantly deteriorates when the parameter's value is small. A plausible explanation for this decline is that the model's optimization process is dominated by the negatives. On the other hand, when the temperature parameter has a larger value, the model also does not perform optimally. This could be attributed to the model requiring a greater number of epochs to converge.
- As the number of devices participating in training increases per round, the model's performance gradually improves and stabilizes. A potential explanation for this improvement is that with a growing number of participating devices, there is a greater probability of incorporating users who actively share their data in each round. This, in turn, enhances the proportion of the model utilizing the contrastive learning strategy, thereby boosting its overall performance.
- The model exhibits optimal performance when the depth of the GNN is set to three layers. A plausible reason for this is that with fewer layers, the GNN may not capture the higher-order graph structural information effectively. On the other hand, when the network is too deep, it might encounter the over-smoothing issue [16], consequently diminishing the model's efficacy.

## 5 CONCLUSION

In this study, we contend that the prevailing FedRecs architecture lacks adaptability and is less enticing. This is primarily because it uniformly assumes a 0-privacy budget for all users. Such an assumption fails to account for those individuals who, being less privacy-conscious, are open to sharing either their complete data or parts of it in exchange for enhanced recommendation services. To address this concern, we delve into a largely untapped area termed as the user-governed data contribution federated recommendation (UGFedRec). This paradigm empowers users with the autonomy to decide if they want to share data and, if so, the extent to which they would share with the server. Building on this concept, we introduce a cloud-device collaborative graph neural network federated recommendation model, dubbed CDCGNNFed. This model facilitates the training of user-centric ego graphs at the local level, while also leveraging high-order graphs constructed from user-contributed data on the server. The collaboration between the two is further enriched through contrastive learning. The efficacy of our proposed approach was validated on two public datasets. The experimental results demonstrate that, within the context of UGFedRec settings, our model consistently outperforms the existing baselines in the vast majority of scenarios. In future work, we intend to explore the integration of our framework with various base recommendation models, such as MF-based recommendation techniques. Additionally, we aim to address the cold-start problem inherent in federated recommender systems.

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
