# OpenReview forum: "Towards Personalized Privacy: User-Governed Data Contribution for Federated Recommendation"
_ACM.org/TheWebConf/2024/Conference — TheWebConf24 Oral_

### Official Review · Reviewer_mJjn · 2023-11-21

**Novelty:** 7
**Technical Quality:** 6

**Review:**

The paper addresses the one-size-fits-all privacy approach in FedRecs by introducing a more flexible, personalized privacy, user-governed framework, allowing users to control the extent of their data shared with the server. The paper introduces the CDCGNNFed model, which combines local training on user-centric ego graphs with server-side training on higher-order graphs, leveraging contrastive learning and a graph mending strategy for improved recommendations. The extensive experiments on public datasets show the effectiveness of the proposed method.

Pros:
1.	The concept of user-governed data contribution in the context of federated recommendation systems is novel and introduces a new perspective on privacy management in FedRecs.
2.	The proposed device-cloud constrastive learning approach, which constructs a positive embedding pair from local embedding and global embedding, makes sense to me.
3.	The experiments are sufficient and demonstrate the superiority of the proposed method compared to existing federated recommendation approaches.
Cons:
1.	Some experimental setups are somewhat confusing.
2.	More technical details about graph mending should be provided.

**Questions:**

1.	Some experimental setups are somewhat confusing. For instance, regarding the top-k experiments with partial uploading, is it that all users sample within the (0,1) interval and then report results of different interval samplings in Tables 2 and 3, or do all users sample within different intervals? The authors should clarify this point.
2.	More technical details about graph mending should be provided. For instance, the GNN-BASED node encoder mentioned in the paper – is it pre-trained and fixed, or is it updated along with the overall federated recommendation model? Clarification on this aspect is needed.

**Ethics Review Description:**

NULL

**Reviewer Confidence:**

4: The reviewer is certain that the evaluation is correct and very familiar with the relevant literature

**Scope:**

4: The work is relevant to the Web and to the track, and is of broad interest to the community

---

### Official Review · Reviewer_ENsd · 2023-11-23

**Novelty:** 7
**Technical Quality:** 6

**Review:**

This paper proposes a novel and interesting federated recommendation setting, called user-governed data contribution federated recommendation (UGFedRec), where users can control whether to share or share part of their data with the server.  Moreover,  the authors present a cloud-device collaborative graph neural network federated recommendation method (CDCGNNFed) to collaboratively train models on the device side and the server side based on local data and data shared from users respectively.  Experimental results on two public datasets demonstrate the effectiveness of the proposed method.

Pros:
1. The motivation behind this work is intriguing and underexplored, providing a fresh perspective for the practical application of federated recommendation systems.
2. The proposed method, i.e., CDCGNNFed, is sound in the context of UGFedRec setting. The use of graph mending to complete graphs is a simple yet logical solution to harness the capability of graph neural networks to capture high-order graph information.
3. The experiments conducted in the paper are comprehensive and serve well to prove the effectiveness of the proposed methods.

Cons:
1. Some of the notations are confusing. If I understand correctly, the regularization coefficient $\lambda$ in Equation (7) should be the same as $\lambda_{2}$ in Equation (8). The authors should use consistent notation.
2. The framework proposed in the paper appears to be model-agnostic, which would make it applicable to other base recommendation methods beyond graph neural networks, such as matrix factorization-based methods. Discussion on this aspect is missing in the paper.

**Questions:**

1. Are the regularization coefficients in Equations (7) and (8) the same? The authors should clarify this.
2. Why don't the authors use traditional matrix factorization methods as the base model? Is the proposed framework applicable to matrix factorization methods?

**Ethics Review Description:**

N.A.

**Reviewer Confidence:**

4: The reviewer is certain that the evaluation is correct and very familiar with the relevant literature

**Scope:**

4: The work is relevant to the Web and to the track, and is of broad interest to the community

---

### Official Review · Reviewer_T8fY · 2023-11-24

**Novelty:** 4
**Technical Quality:** 4

**Review:**

This submission presents a novel approach to address the challenges in federated recommender systems (FedRecs), which have traditionally assumed a uniform privacy policy, where users do not share personal data, leading to a compromise in the system's effectiveness due to data sparsity and lack of personalization. The proposed solution, named CDCGNNFed, introduces a user-governed data contribution model, allowing users to control the extent of their data sharing. This flexibility enhances the recommendation quality while maintaining user privacy preferences.

The approach combines cloud-device collaborative graph neural network models with a graph mending strategy and contrastive learning. This combination effectively leverages the strengths of graph neural networks in handling high-order graph structures and addresses the challenges posed by data sparsity and non-uniform distribution. The experiments conducted on public datasets demonstrate the effectiveness of the proposed method, particularly in scenarios of partial data sharing. The results show some improvement in recommendation quality compared to existing federated recommendation systems and other baselines. The ablation study further highlights the importance of each component in the proposed system, especially the graph mending strategy and the device-cloud contrastive learning component.

Strong points:
Innovative Conceptual Framework: The paper introduces a novel concept in federated recommender systems by allowing user-governed data contribution. This flexibility in data sharing is a significant step forward in addressing privacy concerns in recommender systems.

Robust Methodology: The use of a cloud-device collaborative graph neural network, combined with a graph mending strategy and contrastive learning, is a sophisticated approach that effectively tackles the challenges of data sparsity and non-uniform distribution.

Comprehensive Experiments: The extensive experiments conducted on public datasets provide a solid foundation for the paper's claims. The improvement in recommendation quality over existing systems and baselines is well-documented and convincing.

Relevance and Impact: The topic is highly relevant in today's data-driven world, where privacy concerns are paramount. The proposed model has the potential to influence the development of more user-centric and privacy-aware systems across various sectors.

Presentation: The paper is in general presented well, with a clear structure and logical flow. It effectively communicates complex ideas and methodologies, making it accessible to readers with different levels of expertise.

Reasons to Reject:
User Participation Assumption: The model assumes active user participation in governing their data contribution. This assumption may not hold true in all cases, as users might not always be willing or able to make informed decisions about their data sharing preferences.

Complexity of Implementation: The proposed model's complexity, involving cloud-device collaboration and advanced neural network techniques, might pose challenges in practical implementation, especially in systems with limited resources.

The paper includes several typos: The term “constrastive learning” appears in various sections and should be corrected to “contrastive learning.” In Section 1, the incorrectly spelled word "fliexible" needs to be revised to "flexible," among other similar errors throughout the submission.

I acknowledge that I have read the rebuttal.

**Questions:**

How practical CDCGNNFed for applications with limited computing resources?

If active user participation in governing their data contribution does not hold, how CDCGNNFed would behave?

**Reviewer Confidence:**

4: The reviewer is certain that the evaluation is correct and very familiar with the relevant literature

**Scope:**

4: The work is relevant to the Web and to the track, and is of broad interest to the community

---

### Official Review · Reviewer_r5n1 · 2023-11-27

**Novelty:** 6
**Technical Quality:** 6

**Review:**

This paper explores a user-governed data contribution federated recommendation architecture where users are free to take control of whether they share data and the proportion of data they share with the server. Specifically, this paper presents a cloud-device collaborative graph neural network federated recommendation model, named CDCGNNFed.

**Positive aspects of this paper:**

1. Privacy is always a big concern in designing recommendation algorithms, and thus this paper is well-motivated.
2. The authors have made a rational assumption that users are free to take control of whether they share data and the proportion of data they share with the server.
3. The proposed method can be used in both MF and GNN, which ensures the method can be widely applied.

**Negative aspects of this paper:**
1. The baselines can be enriched. The authors can refer to more papers about controllable privacy/data sharing in the field of recommender systems.
2. The authors can improve the paper presentation further.

**Questions:**

See above.

**Reviewer Confidence:**

4: The reviewer is certain that the evaluation is correct and very familiar with the relevant literature

**Scope:**

4: The work is relevant to the Web and to the track, and is of broad interest to the community

---

### Official Review · Reviewer_qjP8 · 2023-11-30

**Novelty:** 5
**Technical Quality:** 4

**Review:**

The authors present a federated recommendation system based on GNN that attempts to overcome the issue of previous FedRecs, which overlook users less concerned about privacy and willing to share either all or portions of their data to receive a better recommendation service. Specifically, the authors promise to ensure user control over whether they share data and the proportion of data they share with the server.

*Pros:
- The idea is certainly interesting. The assumption of users preferring to share all or part of their data is entirely realistic and deserves consideration.
- The approach regarding graph mending is inspiring in the scenario mentioned above.

*Cons:
- Regarding the Related Work, it would be more useful to delve into the federated recommendation theme, in my opinion, even avoiding presenting the classic centralized RSs. In particular, it would be very helpful to describe and compare works [20] and [21], and, furthermore, explain how other federated works based on GNN perform graph learning and how the proposed approach differs from these.
- The explanation of the approach is not very clear in some cases. For instance, why does e_i-d appear in Equation 4? In this case, it refers to the inference of device-side embeddings. But how does a user possess the embeddings of other users? Perhaps the server also distributes the embeddings of all other users to each user. This aspect is not clearly specified in the approach, and moreover, it doesn't seem optimal for privacy, as it could create a significant information overhead.
- The algorithm should be written by formally presenting the different steps, using mathematical notation and the names of the various calculated/exchanged objects. This could significantly aid in understanding the algorithm and unclear steps.
- How does a user possess the embeddings of other users?
- The calculation of losses in Equations 6 and 7 is performed locally. But are the "local" embeddings those received from the server? So, are those local ones replaced by the global ones in line 11 of the Algorithm?
- Looking at Section 3.5, I understand that the server can calculate the SSL loss, which has all the user embeddings. Why does the algorithm at line 13 say that users locally compute Equation 8, which implicitly requires using SSL loss locally? Once again, the algorithmic steps and the related narrative in Section 3 are unclear.
- Regarding the experimental evaluation: What is meant by "uniform sampling to randomly designate an upload ratio for each user"? The proposed model should be experimented with a portion of users who share nothing and others who share something. In this experiment, with a certain probability, all users share a portion of the data. This would be the proposed novelty, but in my opinion, it is not well tested; instead, a scenario very similar to that of FedeRank is simulated.

Overall, the idea is very interesting, and the proposed approach seems effective. However, the explanation of the approach needs improvement for better clarity and comprehensibility, as well as some experimental comparisons that need to be reviewed.

**Questions:**

- Can you elaborate on how the embedding distribution to the users is managed within the proposed federated recommendation system? Specifically, how does the system address privacy concerns about shared embeddings?
- Could you provide further clarification on how the algorithm performs local and cloud computations and how is the computation of losses managed?
- The experimental evaluation discusses the designation of an upload ratio for each user through uniform sampling. How does this experimental setup effectively test the novelty of the proposed approach, specifically in distinguishing between users who share nothing versus those who share some portion of their data? How could the experimental design be modified to highlight the system's innovation more effectively than similar approaches like FedeRank?

**Ethics Review Description:**

No issue

**Reviewer Confidence:**

4: The reviewer is certain that the evaluation is correct and very familiar with the relevant literature

**Scope:**

3: The work is somewhat relevant to the Web and to the track, and is of narrow interest to a sub-community

---

### Decision · Program_Chairs · 2024-01-22

**Decision:**

Accept (Oral)

**Comment:**

The paper presents a novel and promising approach to federated recommendation systems, introducing a user-governed data contribution model and a cloud-device collaborative graph neural network, named CDCGNNFed. This approach innovatively addresses the one-size-fits-all privacy issue in federated recommendation systems (FedRecs) by allowing users to control the extent of their data sharing. The strength of this paper lies in its innovative conceptual framework, robust methodology, and comprehensive experimental analysis. The paper is well-structured, presenting complex ideas and methodologies clearly and logically. The experiments conducted on public datasets demonstrate the effectiveness of the proposed method, especially in scenarios of partial data sharing, and the results show marked improvement over existing federated recommendation systems. There are areas where the paper could be improved. The explanation of the approach could benefit from greater clarity, and the experimental comparisons need some revision. Additionally, the assumption of active user participation in data sharing decisions and the complexity of the proposed model's implementation could pose challenges in practical scenarios.